# Gastric Emptying of Maltodextrin versus Phytoglycogen Carbohydrate Solutions in Healthy Volunteers: A Quasi-Experimental Study

**DOI:** 10.3390/nu14183676

**Published:** 2022-09-06

**Authors:** Leila R. D. Hammond, Joseph Barfett, Andrew Baker, Néma D. McGlynn

**Affiliations:** 1Enhanced Medical Nutrition, 50 Carroll Street, Toronto, ON M4M 3G3, Canada; 2Department of Medical Imaging, St. Michael’s Hospital, Toronto, ON M5B 1W8, Canada; 3Departments of Critical Care and Anesthesia, St. Michael’s Hospital, Toronto, ON M5B 1W8, Canada

**Keywords:** preoperative carbohydrate, anabolic resistance, carbohydrate loading, insulin, surgical outcomes, enhanced recovery after surgery (ERAS)

## Abstract

Preoperative carbohydrate beverages have been shown to be beneficial in improving patient outcomes. There have been several investigations into the safety of maltodextrin as a preoperative carbohydrate. Although alternative preoperative carbohydrate sources have been proposed, there have been few investigations into the safety and gastric emptying of novel carbohydrate beverages. The present study aimed to compare the gastric emptying of phytoglycogen and maltodextrin to evaluate safety for use as presurgical carbohydrate beverages. In a quasi-experimental design, ten healthy participants orally consumed either a 12.5% maltodextrin or a 12.5% phytoglycogen solution. Gamma scintigraphy was used to evaluate gastric emptying at baseline at 45, 90, and 120 min. Serum insulin and serum glucose were measured at baseline at 15, 30, 45, 60, 90, and 120 min. Gastric volume was significantly lower in the phytoglycogen group at 45 min (*p* = 0.01) and 90 min (*p* = 0.01), but this difference lost significance at 120 min (*p* = 0.17). There were no significant differences between treatments for serum insulin or serum glucose at any time point. This study indicates that the gastric emptying of phytoglycogen is comparable to maltodextrin at 120 min after ingestion, opening the opportunity for the study of alternative carbohydrates for utilization as preoperative carbohydrates.

## 1. Introduction

The use of complex carbohydrate beverages preoperatively is becoming increasingly common in the pursuit of improved patient outcomes. Several Enhanced Recovery After Surgery (ERAS) guidelines recommend preoperative carbohydrate loading in the form of a 12.5% carbohydrate solution [1,2]. This form of preoperative carbohydrate loading improves perioperative patient outcomes by attenuating post-surgical catabolism and improving patient discomfort brought on by surgical stress [2,3,4,5]. Specifically, preoperative complex carbohydrate loading produces an insulin response to prevent insulin resistance after surgery, which is caused by the additive effects of fasting and surgical stress [2,3,4,5,6]. Maltodextrin is the most commonly used carbohydrate in preoperative carbohydrate beverages [3]. While maltodextrin is advantageous as a carbohydrate due to its relatively low osmolality in solution and ability to produce a strong glycemic and insulinemic response, it is not the only carbohydrate with these qualities [3].

Alternatives for maltodextrin are high molecular weight (HMW) carbohydrates. These polysaccharides have been studied for their effect on fuel availability and glycogen synthesis in the context of athletic performance [7,8,9,10]. The same characteristics that make them effective in a sports performance context may also make them effective for preoperative carbohydrate loading. HMW carbohydrates have lower osmolality, faster gastric emptying, produce greater rates of glycogen synthesis, and have similar glycemic and insulinemic responses compared with maltodextrin and other low molecular mass carbohydrates [7,8,9,10]. One such HMW carbohydrate is phytoglycogen, a corn-derived, highlybranched carbohydrate. Phytoglycogen’s unique structure may confer improved glycemic and insulinemic responses perioperatively [11,12]. Despite its potential benefit to perioperative outcomes, no trials have evaluated phytoglycogen for application in this setting.

Several surgical and anesthesia societies have recently recommended the ingestion of clear fluids up to two hours before surgery in adults with normal gastric motility [5,13,14]. These recommendations are based on evidence demonstrating that clear fluid safely empties from the stomach within two hours of ingestion with no increased risk of aspiration [13]. Therefore, the aim of the present study was to evaluate the gastric emptying of two presurgical carbohydrate beverages—a standard maltodextrin solution and a novel phytoglycogen solution—over the course of 2 h in adults.

## 2. Materials and Methods

### 2.1. Participants

In a quasi-experimental design, healthy volunteers were recruited by word-of-mouth. Inclusion criteria were healthy adults aged 20–50 years. Exclusion criteria were a history of gastric surgery, taking any medications known to alter gastric motility, pregnancy, class 2 obesity or higher (BMI > 35 kg/m^2^), type 1 or type 2 diabetes mellitus, and any gastrointestinal disease. Eligibility was determined by initial telephone screening. Written consent was obtained on the study’s test day by the nuclear technologist. Twelve participants were recruited, and two were excluded due to deviations from the study protocol. Participants were placed into one of two intervention groups: a maltodextrin solution (MTD) group or a phytoglycogen solution group (PGN). Group allocation occurred such that the first 7 participants were assigned MTD, and the subsequent 5 participants were assigned PGN for convenience. A total of 10 participants were included in the final analysis. The current study was approved by the Unity Health Research Ethics Board (REB-18-080).

### 2.2. Study Beverages

The maltodextrin solution was constituted from 54 g of a commercially available citrus-flavored preoperative carbohydrate powder (PREcovery, Enhanced Medical Nutrition, Toronto, ON, Canada) mixed into 400 mL of water to form a 12.5% (*w*/*v*) maltodextrin solution providing 50 g of maltodextrin. The phytoglcogen solution was developed for this study and matched the composition of the maltodextrin solution, except for the carbohydrate source, which was phytoglycogen (Mirexus, Guelph, ON, Canada). The phytoglycogen solution was mixed with 400 mL of water to provide a 12.5% (*w*/*v*) phytoglycogen solution, providing 50 g of carbohydrates. Both carbohydrate solutions were mixed with Tc99m sulfur colloid to a concentration of 37 MBq +/− 10%, suitable for detection using the gamma scintigraphy methods described.

### 2.3. Test Day

Participants arrived at the testing facility at 08:30. They were instructed to fast 6 h prior to the scheduled test initiation time, in accordance with the presurgical fasting guidance from several anesthesia societies [13,14]. Participants’ weight, height, sex, and date of birth were collected upon arrival. At 09:00, participants were instructed to drink their respective study beverage within 5 min.

### 2.4. Gamma Scintigraphy

A dual-head gamma camera (Infinia Hawkeye 4, GE Healthcare, Chicago, IL, USA) was used to assess gastric emptying. Participants stood between the two detector heads, and the camera was placed at the minimal distance thatallowed it to navigate body habitus. The detectors had low-energy and high-resolution collimators suitable for the 37 MBq +/− 10% Tc99m sulfur colloid. The participants’ stomach was centered in the field of view. The camera was placed in H mode (upright) position to acquire simultaneous anterior and posterior images of the participants’ thorax and upper abdomen. Digital image and communication data were processed and stored locally on a workstation. Images were taken at 45, 90, and 120 min after ingestion of the maltodextrin or phytoglycogen solutions. Gastric emptying was analyzed as the initial amount of radioactivity at baseline less than the amount of radioactivity remaining in the stomach at the given timepoint, as a percentage of the initial radioactivity.

### 2.5. Blood Samples

Venous blood samples were collected at baseline and 15, 30, 45, 60, 90, 120, 150, and 180 min after consuming the respective carbohydrate solution. Blood samples were collected by inserting an angiocatheter into a vein in the participants’ antecubital fossa using a standard aseptic technique. Serum samples were obtained by allowing blood samples to clot and centrifugation at 3000 rpm for 7 min. Serum samples were analyzed at St. Michaels Hospital, Laboratory Medicine Department, Toronto, ON, Canada. Serum glucose concentrations were measured using the glucose hexokinase method. Serum insulin was measured using an enzyme immunoassay kit (Mercodia AB insulin, Uppsala, Sweden).

### 2.6. Statistical Analysis

Data are reported as mean ± standard error of the mean (SEM). Statistical significance was set at *p* < 0.05. The incremental area under the curve (iAUC_60_) for insulin and glucose was calculated using the trapezoid method for the first 60 min after ingestion of the respective study beverages. Serum insulin and glucose analyses were completed as a change from baseline measurement. All statistical analyses were performed using the R 4.0.5 statistical package. Mann–Whitney U tests were performed to compare between-group differences for all measures.

## 3. Results

### 3.1. Participants

Groups had an equal number of participants (MTD: n = 5, PGN: n = 5). There were no statistically significant differences between MLT and PGN groups for age or body mass index (BMI) (Table 1).

### 3.2. Gastric Emptying

Those in the PGN group had greater emptying of gastric contents than MTD at 45 (*p* = 0.01) and 90 (*p* = 0.01) minutes; however, this difference was no longer significant at 120 min (*p* = 0.17) (Table 2).

### 3.3. Blood Measures

Differences between MTD and PGN exchanges from baseline serum glucose, and serum insulin are represented in Figure 1 and Figure 2, respectively, and in Table 3. Significance was approached for serum glucose change from baseline at 30 min (*p* = 0.06) and 45 min (*p* = 0.09). Similarly, serum insulin change from baseline values approached significance at 30 min (*p* = 0.06). The mean iAUC_60_ for serum glucose was 60.4 mmol/L (±14.82 SEM) and 105.1 mmol/L (± 10.11 SEM) for MTD and PGN, respectively (*p* = 0.06); the mean glucose iAUC_60_ was 73.8% larger for the PGN group than the MTD group. The mean iAUC_60_ for serum insulin was 7428 mmol/L (±936.7 SEM) and 16830 mmol/L (±6635.9 SEM) for MTD and PGN, respectively (*p* = 0.14); the mean insulin iAUC_60_ was 126% greater in the PGN group than the mean insulin iAUC_60_ in the MTD group.

## 4. Discussion

To our knowledge, this study is the first to evaluate phytoglycogen compared to maltodextrin for its effects on human gastric emptying. The present study found that a 12.5% phytoglycogen solution emptied faster from the stomach than a 12.5% maltodextrin solution at 45 and 90 min after ingestion, but the difference lost significance at 120 min. These findings are congruent with studies comparing the gastric emptying of other HMW carbohydrates and maltodextrin, where HMW carbohydrates have increased rates of gastric emptying than lower molecular weight carbohydrates [15]. While both maltodextrin and phytoglycogen are polysaccharides derived from corn, maltodextrin has a short chain length and a molecular mass ranging from 0.5 to 2.8 kDa [16]. In contrast, phytoglycogen has a highly branched, dendrimeric structure and high molecular weight at 500–10,000 kDa [11]. Correspondingly, phytoglycogen has a lower osmolality in solution than maltodextrin while retaining high water retention and low viscosity, which may have produced the expedited gastric emptying seen in the present study [12]. The mechanism for this increased rate of gastric emptying may be due to PGN’s comparatively low osmolality due to its high molecular weight [15,17], as influences gastric motility [17,18].

This present study adds to the literature demonstrating that a 12.5% maltodextrin solution safely empties from the stomach within 2 h [4,13,19]. Previous research has established 1.5 mL/kg body weight as the gastric volume threshold that does not increase the risk of aspiration [20]. In the present study, all but one participant in the MTD group had gastric contents of <1.5 mL/kg at 120 minutes, which is consistent with the rate found in other studies [19,21]. Interestingly, all participants in the PGN group had gastric contents <1.5 mL/kg at 120 min, although gastric emptying was not significantly lower than the MTD group at this time point. This finding suggests that other carbohydrate solutions may be useful for preoperative carbohydrate loading within a shorter time period before surgery or for individuals with delayed gastric emptying. This finding is especially pertinent as there are emerging calls for the fasting window to be shortened further to 1 h preoperatively [22].

Alhough the literature on the topic is equivocal, some studies observed that the ingestion of oral fluids results in lesser gastric volumes than fasting, though this seems dependent on the type and volume of fluid administered and the amount of time before measurement [8,23,24,25,26]. Lesser gasteic volumes after fluid ingestion may be due to ingested fluids stimulating the gastric emptying of all gastric contents (i.e., gastric fluid and ingested fluid) at a greater rate than the gastric emptying of gastric fluid only. In the present study, the mean values for gastric emptying of the MTD and PGN groups were 83.8% and 95.6%, respectively. It is unclear whether the radioactive agent used in the present trial dissolved uniformly in gastric fluids in the stomach at the time of the ingestion of the study’s beverages or during gastric secretions that took place over the gastric emptying study. Considering the properties of the radioactive agent used, it is likely that the gastric emptying percentage values represent not only the emptying of the carbohydrate beverage that was consumed, but also the emptying of some gastric secretions that were in the stomach at the time of ingestion of the carbohydrate beverage [27]. Typical gastric volumes in fasting participants are about 35 mL [28,29]. Considering this, in the present study, gastric volumes at 120 min after the ingestion of 400 mL of a 12.5% carbohydrate solution, may be less than the average fasted gastric volumes reported in the literature.

The present study found that there were no significant differences in blood glucose or insulin responses between the two groups. Despite this, significance was approached for serum glucose change from baseline at 30 min and 45 min, serum insulin change from baseline at 30 min, and both serum insulin and serum glucose iAUC_60_. While the present study was not powered to detect such differences, these results indicate that respective carbohydrate solutions may produce differences in glycemia and insulinemia. The trend toward increased glycemic and insulinemic responses in the PGN group may be influenced by the increased rate of gastric emptying observed at 45 and 90 min, as gastric emptying mediates post-prandial glycemic responses [30,31]. No other studies have specifically compared phytoglycogen with maltodextrin for their effects on glycemic and insulinemic responses. However, Stephens et al. [8] found that a 10% HMW carbohydrate solution providing 100 g of carbohydrate resulted in quicker and greater increases in blood glucose and serum insulin than a matched low molecular mass carbohydrate after glycogen depletion from exercise. Conversely, most other studies comparing maltodextrin with HMW carbohydrates have not found a difference in insulinemia or glycemia [7,9,10,15,17,32]. All but one of the previous studies comparingcarbohydrates of different molecular weights for their effects on blood glucose and insulin occurred after exercise depletions of glycogen [7,9,10,15,17,32], which may have altered glucoregulatory responses in these studies due to non-insulin-mediated glucose transportation potentiated by physical activity.

Only one other human study examined the effects of ingested phytoglycogen [11]. Bandegan et al. [11] found that at different phytoglycogen concentrations in solution, blood glucose levels peaked at 15 min post ingestion of the solution; blood insulin responses to phytoglycogen were not measured. In the present study, blood glucose peaked 30 min after consumng the carbohydrate solution in both the MTD and PGN groups. Bandegan et al. [11] also found that glycemic responses were similar with phytoglycogen concentrations of 7.2%, 10.8%, and 14.4%. The present study used a phytoglycogen concentration of 12.5%. Although there is no consensus on the insulin response required to attenuate catabolism in the postoperative period, these results may indicate that the optimal carbohydrate concentration required to produce a substantial glycemic response, that elicits a clinically significant insulin response, may differ depending on the properties of the carbohydrate used [6].

While the present study includes important insights on the gastric emptying kinetics of phytoglycogen and maltodextrin, it has some limitations. The present study was not adequately powered to detect differences in serum glycemic and insulinemic responses. Therefore, research powered for those outcomes should be conducted to properly elucidate the differences in glucoregulatory responses to the respective carbohydrate solutions. The present study used healthy participants as a proxy for surgical patients, which may limit the generalizability of these results to surgical patients. Several factors unique to the surgical setting (e.g., anxiety, pain, and pharmacological agents), in addition to any pathophysiology related or unrelated to the indication for surgery, may influence gastric emptying and glucoregulatory responses [5,33]. Despite updated anesthesia guidelines that allow the ingestion of clear fluids up to 2 h before surgery, more research is necessary to replicate the present study’s findings in different surgical populations. Future research should also measure postoperative outcomes in various surgical populations, especially outcomes related to insulin sensitivity and catabolism. 

Furthermore, gamma scintigraphy has limitations for measuring gastric contents.It does not allow for baseline gastric content measurements, and in the context of liquid gastric emptying, it is unclear if the radioisotope tracer dissolves into the gastric fluid to capture the total gastric volume. For the sake of convenience, the participants in this study were not randomized such that the first five participants were in the MTD group, and the following five were in the PGN group, increasing the risk of bias in the data. More high-quality, randomized trials are needed in this area. Future research on the impact of carbohydrate type in the context of presurgical carbohydrate beverages on perioperative patient outcomes to inform optimal perioperative care is needed.

In conclusion, both phytoglycogen and maltodextrin solutions demonstrate excellent gastric emptying within 120 min. In addition, the respective carbohydrates may elicit slight differences in insulin and glycemic responses, which may inform future research on presurgical carbohydrate loading.

## Figures and Tables

**Figure 1 nutrients-14-03676-f001:**
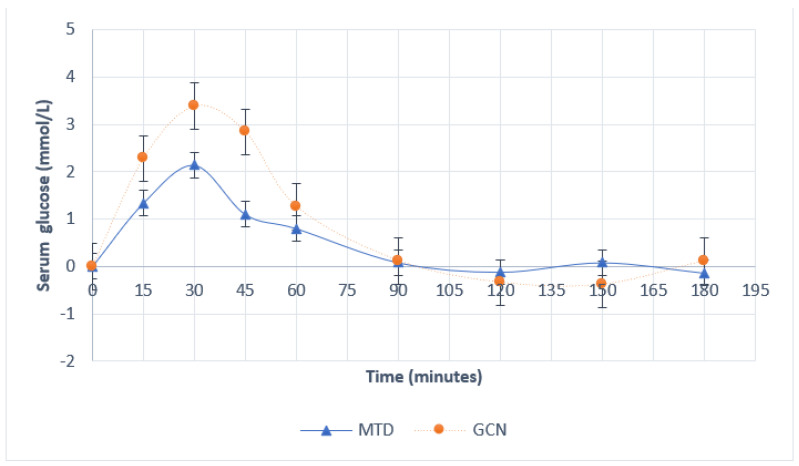
Serum glucose concentrations over time. Error bars represent SEM.

**Figure 2 nutrients-14-03676-f002:**
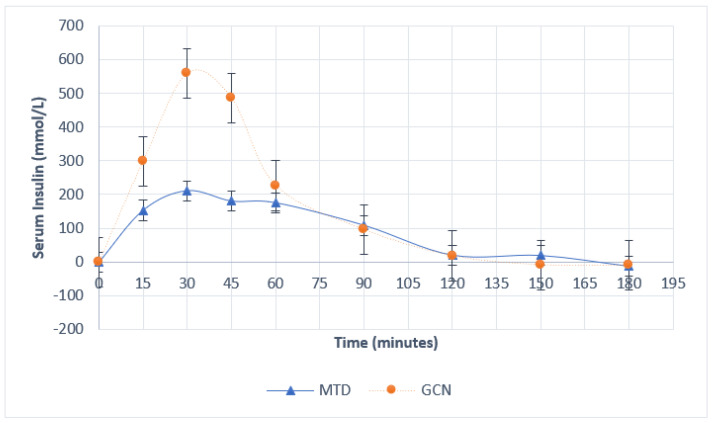
Serum insulin concentrations over time. Error bars represent SEM.

**Table 1 nutrients-14-03676-t001:** Participant Characteristics.

Characteristic	Intervention Group		*p*-Value
	Maltodextrin	Phytoglycogen	
n	5	5	
Age (years)	27.1 ± 1.66	24.8 ± 0.86	0.40
BMI (kg/m^2^)	24.9 ± 2.32	25.6 ± 1.80	1
Sex (male/female)	3/2	3/2	

Values are expressed as mean ± SEM. Mean participant characteristics for Age, BMI, and sex in respective MTD and PGN groups. The *p*-values were calculated via Mann–Whitney U tests. BMI = body mass index.

**Table 2 nutrients-14-03676-t002:** Gastric Emptying.

Time Point	%Gastric Emptying		*p*-Value
	Intervention Group		
	Maltodextrin	Phytoglycogen	
45 min	38.6 ± 6.06	68.0 ± 3.24	0.01
90 min	66.8 ± 5.57	87.4 ± 0.96	0.01
120 min	83.8 ± 6.07	95.6 ± 0.88	0.17

Values are expressed as mean ± SEM. Gastric emptying at specific time points is expressed as the initial amount of radioactivity at baseline less than the amount of radioactivity remaining in the stomach at the given timepoint, as a percentage of the initial radioactivity.

**Table 3 nutrients-14-03676-t003:** Serum Glucose and Serum Insulin.

Time Point (Minutes)	Serum Glucose (mmol/L)					Serum Insulin (mmol/L)				
	Intervention Group				*p*-Value	Intervention Group				*p*-Value
	Maltodextrin		Phytoglycogen			Maltodextrin		Phytoglycogen		
15	1.34	±0.48	2.28	±0.22	0.29	153.0	±50.53	298.6	±114.91	0.83
30	2.14	±0.38	3.38	±0.25	0.06	210.6	±34.38	559.6	±207.53	0.06
45	1.10	±0.54	2.84	±0.43	0.09	180.0	±25.43	486.8	±217.50	0.21
60	0.80	±0.55	1.26	±0.33	0.67	175.0	±33.20	226.6	±78.86	0.83
90	0.08	±0.46	0.12	±0.29	1	108.6	±33.02	95.8	±20.43	0.83
120	−0.12	±0.45	−0.34	±0.31	0.91	20.6	±27.8	18.2	±11.18	0.83
150	0.08	±0.45	−0.38	±0.20	0.67	19.6	±25.69	−8.8	±8.48	1
180	−0.14	±0.33	0.12	±0.10	0.24	−12.0	±12.95	−10.4	±6.84	1

Values are expressed as mean ± SEM. Changes from baseline serum glucose and serum insulin are reported.

## Data Availability

The data presented in this study are available upon request from the corresponding author.

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
