# Peer review of "Gastric Emptying of Maltodextrin versus Phytoglycogen Carbohydrate Solutions in Healthy Volunteers: A Quasi-Experimental Study"

_nutrients, 2022, doi:10.3390/nu14183676_

Round 1
Reviewer 1 Report
Gastric Emptying of Maltodextrin versus Phytoglycogen Carbo-2 hydrate Solutions in Healthy Volunteers.
Leila Hammond 1, *, Joseph Barfett 2, Andrew Baker 3 and Néma McGlynn 1
Hammond et al present an important study wherein they have compared Maltodextrin versus Phytoglycogen carbohydrate solution for gastric emptying. While the transparency with which the authors have discussed the study's limitations is appreciable, however, it is sad that the study was not adequately powered to draw any conclusions about many of the observations. In the absence of adequate power, this study has limited scope for interpretation.
Specific comments:
1) Phytoglycogen carbohydrate ingestion seems to increase the rate of gastric emptying, which can cause postprandial hyperglycaemia (Goyal, Guo, and Mashimo 2019) (as also seen in this study), please discuss its implications.
2) Plasma glucose AUC increased by 74% (105.1 vs 60.4) in the PGN group when compared to the MTD group, it should not be called a non-significant difference when the study is not powered to test significance.
3) Similarly, plasma insulin AUC increased by >125%, and hence should be concluded as significant or nonsignificant unless the study is adequately powered to make such conclusions. please rephrase it.
4) Please discuss the trend toward increased glucose and insulin in the context of diabetic individuals.
Goyal, Raj K., Yanmei Guo, and Hiroshi Mashimo. 2019. “Advances in the Physiology of Gastric Emptying.” Neurogastroenterology and Motility: The Official Journal of the European Gastrointestinal Motility Society 31 (4): e13546.
Author Response
Response to reviewer 1
Hammond et al present an important study wherein they have compared Maltodextrin versus Phytoglycogen carbohydrate solution for gastric emptying. While the transparency with which the authors have discussed the study's limitations is appreciable, however, it is sad that the study was not adequately powered to draw any conclusions about many of the observations. In the absence of adequate power, this study has limited scope for interpretation.
Thank you for your detailed review of the manuscript and thoughtful and thorough responses. I have addressed your comments below and in the manuscript. Edits to the manuscript have been done through track changes and I have referenced the corresponding lines in the manuscript in the respective responses.
We agree with this appraisal and are disappointed with our inability to adequately power this study for the blood measures. Unfortunately, the design and sample for this study had to be altered due to hospital visitor restrictions and resource reallocation caused by the COVID-19 pandemic. Several restrictions remain at the institution where this study took place, and it is not possible to include more participants at this time; we are unsure of when St. Michaels hospital will remove restrictions for their research activities. While the limited power of these measures diminishes the impact of the study, we believe that this study is a valuable contribution to the literature on the topic of gastric emptying, presurgical carbohydrate drinks and could be used to inform future research and guidance on these important topics. This study is the first to evaluate the impact of carbohydrate structure within the context of presurgical carbohydrate drinks. There are several guidance documents that recommend the use of maltodextrin for pre-surgical applications though there is no evidence to support this, and other guidance do not specify carbohydrate type at all (1–3). This study may inform a new area of research on the impact of carbohydrate structure and the use of novel carbohydrates for improving patient outcomes and producing better evidence informed guidance.
- Phytoglycogen carbohydrate ingestion seems to increase the rate of gastric emptying, which can cause postprandial hyperglycaemia (Goyal, Guo, and Mashimo 2019)(as also seen in this study), please discuss its implications.
- Thank you for providing this pertinent reference. After carefully reviewing this comprehensive review of gastric emptying, I have included built on the discussion of blood glucose response to reflect the advancements in the area of glycemic response’s effect on gastric emptying. Please see lines 221 to 223.
- Plasma glucose AUC increased by 74% (105.1 vs 60.4) in the PGN group when compared to the MTD group, it should not be called a non-significant difference when the study is not powered to test significance.
- I have adjusted the wording to better reflect the serum glucose AUC mean differences between the MTD and PGN group in light of the statistical power of this study. See lines 149, 151-152 and table 3.
- Similarly, plasma insulin AUC increased by >125%, and hence should be concluded as significant or nonsignificant unless the study is adequately powered to make such conclusions. please rephrase it.
- I have adjusted the wording to better reflect the serum insulin AUC mean differences between the MTD and PGN group in light of the statistical power of this study. See lines 156-157 and table 3
- Please discuss the trend toward increased glucose and insulin in the context of diabetic individuals.
- Diabetes was not mentioned in this article as it is beyond the scope of this article as current guidance does not support the use of presurgical carbohydrates in this population. There is limited evidence for the use of presurgical carbohydrate drinks in a diabetic population and opinions on the use of presurgical carbohydrates are equivocal. Therefore, the use of presurgical carbohydrates in this population is not common. Further research is necessary to evaluate the benefits of presurgical carbohydrate drinks in this population, especially with the use of novel carbohydrates.
Goyal, Raj K., Yanmei Guo, and Hiroshi Mashimo. 2019. “Advances in the Physiology of Gastric Emptying.” Neurogastroenterology and Motility: The Official Journal of the European Gastrointestinal Motility Society 31 (4): e13546.
- Pogatschnik C, Steiger E. Review of Preoperative Carbohydrate Loading. Nutr Clin Pract [Internet]. 2015 Oct 8 [cited 2022 Mar 28];30(5):660–4. Available from: https://pubmed.ncbi.nlm.nih.gov/26197803/
- Feldheiser A, Aziz O, Baldini G, Cox BPBW, Fearon KCH, Feldman LS, et al. Enhanced Recovery After Surgery (ERAS) for gastrointestinal surgery, part 2: consensus statement for anaesthesia practice. Acta Anaesthesiol Scand [Internet]. 2016 Mar 1 [cited 2022 Mar 28];60(3):289–334. Available from: https://onlinelibrary.wiley.com/doi/full/10.1111/aas.12651
- Melloul E, Lassen K, Roulin D, Grass F, Perinel J, Adham M, et al. Guidelines for Perioperative Care for Pancreatoduodenectomy: Enhanced Recovery After Surgery (ERAS) Recommendations 2019. World J Surg. 2020 Jul 1;44(7):2056–84.

Reviewer 2 Report
This study was designed to evaluate the rates of gastric emptying of two presurgical carbohydrate beverages, a standard maltodextrin solution and a novel phytoglycogen solution in 10 healthy adults (five for the maltodextrin and five for the phytoglycogen experiments). The results showed that gastric emptying was faster after phytoglycogen than after maltodextrin administration. However, post-load serum glucose and insulin levels were not significantly different between treatments.
This is a well-presented manuscript dealing with an interesting topic. Prior to surgery, the body gradually enters a catabolic state and carbohydrate loading with an appropriate beverage is crucial to fill glycogen stores in liver and muscle. I have a few comments:
(1) The finding that the rates of gastric emptying were increased more after phytoglycogen than maltodextrin is interesting and clinically useful. However, the clinical importance of such an effect should be supported by post-load increases in serum glucose and insulin levels. In this study, although there was a clear trend for serum glucose and insulin to increase more after phytoglycogen than maltodextrin, the differences did not reach statistical significance due to the small number of subjects. If the authors add 2-3 more subjects to each group, they will easily get statistical significance in the glucose and insulin data and support the clinical usefulness of the treatment.
(2) Why did the authors instruct their subjects to fast only for 4-6 hours prior to scheduled initiation time? Following meals, the increases in plasma insulin levels usually require more than six hours to return to baseline and insulin action persists even more than that. Therefore, prior to metabolic experiments, the usual fasting period is 10-12 hours to ensure a steady state. The lack of a steady state may be one of the reasons why there were no statistically significant differences in insulin and glucose between treatments although the trend was clear.
(3) In the Results section and Table 2, I see that the subjects in the phytoglycogen group had greater emptying of gastric contents than maltodextrin at 45 and 90 minutes. However, in the Abstract (lines 22-23) and Conclusions (lines 251-252) I see that “the gastric emptying of phytoglycogen is comparable to maltodextrin”. Please clarify.
Author Response
Response to Reviewer 2
This study was designed to evaluate the rates of gastric emptying of two presurgical carbohydrate beverages, a standard maltodextrin solution and a novel phytoglycogen solution in 10 healthy adults (five for the maltodextrin and five for the phytoglycogen experiments). The results showed that gastric emptying was faster after phytoglycogen than after maltodextrin administration. However, post-load serum glucose and insulin levels were not significantly different between treatments.
This is a well-presented manuscript dealing with an interesting topic. Prior to surgery, the body gradually enters a catabolic state and carbohydrate loading with an appropriate beverage is crucial to fill glycogen stores in liver and muscle. I have a few comments:
Thank you for your detailed review of the manuscript and thoughtful and thorough responses. I have addressed your comments below and in the manuscript. Edits to the manuscript have been done through track changes and I have referenced the corresponding lines in the manuscript in the respective responses.
- The finding that the rates of gastric emptying were increased more after phytoglycogen than maltodextrin is interesting and clinically useful. However, the clinical importance of such an effect should be supported by post-load increases in serum glucose and insulin levels. In this study, although there was a clear trend for serum glucose and insulin to increase more after phytoglycogen than maltodextrin, the differences did not reach statistical significance due to the small number of subjects. If the authors add 2-3 more subjects to each group, they will easily get statistical significance in the glucose and insulin data and support the clinical usefulness of the treatment.
- Thank you for your commentary. We agree with this appraisal and are disappointed with our inability to adequately power this study for the blood measures. Unfortunately, the design and sample for this study had to be altered due to hospital visitor restrictions and resource reallocation caused by the COVID-19 pandemic. Several restrictions remain at the institution where this study took place, and it is not possible to include more participants at this time; we are unsure of when St. Michaels hospital will remove restrictions for their research activities. While the limited power of these measures diminishes the impact of the study, we believe that this study is a valuable contribution to the literature on the topic of gastric emptying, presurgical carbohydrate drinks and could be used to inform future research and guidance on these important topics. This study is the first to evaluate the impact of carbohydrate structure within the context of presurgical carbohydrate drinks. There are several guidance documents that recommend the use of maltodextrin for pre-surgical applications though there is no evidence to support this, and other guidance do not specify carbohydrate type at all (Feldheiser et al., 2016; Melloul et al., 2020; Pogatschnik & Steiger, 2015). This study may inform a new area of research on the impact of carbohydrate structure and the use of novel carbohydrates for improving patient outcomes and producing better evidence informed guidance.
- Why did the authors instruct their subjects to fast only for 4-6 hours prior to scheduled initiation time? Following meals, the increases in plasma insulin levels usually require more than six hours to return to baseline and insulin action persists even more than that. Therefore, prior to metabolic experiments, the usual fasting period is 10-12 hours to ensure a steady state. The lack of a steady state may be one of the reasons why there were no statistically significant differences in insulin and glucose between treatments although the trend was clear.
- Referencing participant instructions, subjects were instructed to fast for 6 hours before (the 4-6 hours was what was reported in protocol, but the participant instruction document, the consent form and REB document reflect only 6 hours. I am not sure the reason for this discrepancy), this has been amended in line 86-88. The fasting and preparation instructions for the present study were intended to be ecologically relevant for presurgical instructions that comply with the most recent anesthesia and surgery guidelines. Current American Society of Anesthesiologists, Canadian Anesthesiologists' Society and others guidelines allow for the consumption of solid food up to 6 hours before surgery (Dobson et al., 2021; Sharma et al., 2018). The European Society of Anesthesiology and Intensive Care allow the consumption of solid food and milk up to 6 hours before surgery(Smith et al., 2011). Relatedly, participants were asked to arrive in the morning at 08h30 therefore it is likely that patients were fasted overnight upon arrival. Time of last meal was not recorded, but patient compliance to the 6 hour fasting instruction was. Baseline serum glucose measures were consistent with what would be expected for healthy fasted individuals.
- In the Results section and Table 2, I see that the subjects in the phytoglycogen group had greater emptying of gastric contents than maltodextrin at 45 and 90 minutes. However, in the Abstract (lines 22-23) and Conclusions (lines 251-252) I see that “the gastric emptying of phytoglycogen is comparable to maltodextrin”. Please clarify.
- I have amended the manuscript to be clearer on this finding, see line 23. The 120-minute timepoint was what was being referred to in the abstract and the conclusion as presurgical carbohydrates are typically administered 2 hours before surgery and the most relevant marker for safety would be the gastric contents at 120 minutes.
Dobson, G., Chow, L., Filteau, L., Hurdle, H., McIntyre, I., Milne, A., Milkovich, R., Perrault, M. A., Sparrow, K., Swart, P. A., & Wang, Y. (2021). Guidelines to the Practice of Anesthesia – Revised Edition 2021. Canadian Journal of Anesthesia, 68(1), 92–129. https://doi.org/10.1007/s12630-020-01842-x
Feldheiser, A., Aziz, O., Baldini, G., Cox, B. P. B. W., Fearon, K. C. H., Feldman, L. S., Gan, T. J., Kennedy, R. H., Ljungqvist, O., Lobo, D. N., Miller, T., Radtke, F. F., Ruiz Garces, T., Schricker, T., Scott, M. J., Thacker, J. K., Ytrebø, L. M., & Carli, F. (2016). Enhanced Recovery After Surgery (ERAS) for gastrointestinal surgery, part 2: consensus statement for anaesthesia practice. Acta Anaesthesiologica Scandinavica, 60(3), 289–334. https://doi.org/10.1111/AAS.12651
Melloul, E., Lassen, K., Roulin, D., Grass, F., Perinel, J., Adham, M., Wellge, E. B., Kunzler, F., Besselink, M. G., Asbun, H., Scott, M. J., Dejong, C. H. C., Vrochides, D., Aloia, T., Izbicki, J. R., & Demartines, N. (2020). Guidelines for Perioperative Care for Pancreatoduodenectomy: Enhanced Recovery After Surgery (ERAS) Recommendations 2019. World Journal of Surgery, 44(7), 2056–2084. https://doi.org/10.1007/S00268-020-05462-W
Pogatschnik, C., & Steiger, E. (2015). Review of Preoperative Carbohydrate Loading. Nutrition in Clinical Practice : Official Publication of the American Society for Parenteral and Enteral Nutrition, 30(5), 660–664. https://doi.org/10.1177/0884533615594013
Sharma, S., Deo, A. S., & Raman, P. (2018). Effectiveness of standard fasting guidelines as assessed by gastric ultrasound examination: A clinical audit. Indian Journal of Anaesthesia, 62(10), 747. https://doi.org/10.4103/IJA.IJA_54_18
Smith, I., Kranke, P., Murat, I., Smith, A., O’Sullivan, G., Søreide, E., Spies, C., & in’t Veld, B. (2011). Perioperative fasting in adults and children: guidelines from the European Society of Anaesthesiology. European Journal of Anaesthesiology, 28(8), 556–569. https://doi.org/10.1097/EJA.0B013E3283495BA1

Round 2
Reviewer 2 Report
The authors have adequately addressed the comments raised by this reviewer. I have no further comments.